# Expression of Periostin Alternative Splicing Variants in Normal Tissue and Breast Cancer

**DOI:** 10.3390/biom14091093

**Published:** 2024-08-31

**Authors:** Yuko Kanemoto, Fumihiro Sanada, Kana Shibata, Yasuo Tsunetoshi, Naruto Katsuragi, Nobutaka Koibuchi, Tetsuhiro Yoshinami, Koichi Yamamoto, Ryuichi Morishita, Yoshiaki Taniyama, Kenzo Shimazu

**Affiliations:** 1Department of Breast and Endocrine Surgery, Osaka University Graduate School of Medicine, Suita 565-0871, Japan; y.kanemoto@onsurg.med.osaka-u.ac.jp (Y.K.); yosinami-te@onsurg.med.osaka-u.ac.jp (T.Y.); 2Department of Clinical Gene Therapy, Osaka University Graduate School of Medicine, Suita 565-0871, Japan; sanada@cgt.med.osaka-u.ac.jp (F.S.); tsunetoshi@geriat.med.osaka-u.ac.jp (Y.T.); katsuragi@cgt.med.osaka-u.ac.jp (N.K.); koibuchi@cgt.med.osaka-u.ac.jp (N.K.); morishit@cgt.med.osaka-u.ac.jp (R.M.); 3Department of Advanced Molecular Therapy, Osaka University Graduate School of Medicine, Suita 565-0871, Japantaniyama@cgt.med.osaka-u.ac.jp (Y.T.); 4Department of Geriatric and General Medicine, Osaka University Graduate School of Medicine, Suita 565-0871, Japan; kyamamoto@geriat.med.osaka-u.ac.jp

**Keywords:** periostin, extracellular matrix protein, alternative splicing variants, breast cancer

## Abstract

(1) Background: Periostin (Pn) is a secreted protein found in the extracellular matrix, and it plays a variety of roles in the human body. Physiologically, Pn has a variety of functions, including bone formation and wound healing. However, it has been implicated in the pathogenesis of various malignant tumors and chronic inflammatory diseases. Pn has alternative splicing variants (ASVs), and our previous research revealed that aberrant ASVs contribute to the pathogenesis of breast cancer and heart failure. However, the difference in expression pattern between physiologically expressed Pn-ASVs and those expressed during pathogenesis is not clear. (2) Methods and results: We examined normal and breast cancer tissues, focusing on the Pn-ASVs expression pattern to assess the significance of pathologically expressed Pn-ASVs as potential diagnostic and therapeutic targets. We found that most physiologically expressed Pn isoforms lacked exon 17 and 21. Next, we used human breast cancer and normal adjacent tissue (NAT) to investigate the expression pattern of Pn-ASVs under pathological conditions. Pn-ASVs with exon 21 were significantly increased in tumor tissues compared with NAT. In situ hybridization identified the synthesis of Pn-ASVs with exon 21 in peri-tumoral stromal cells. Additionally, the in vivo bio-distribution of ^89^Zr-labeled Pn antibody against exon 21 (Pn-21Ab) in mice bearing breast cancer demonstrated selective and specific accumulation in tumors, while Pn-21Ab significantly suppressed tumor growth in the mouse breast cancer model. (3) Conclusions: Together, these data indicate that Pn-ASVs might have potential for use as diagnostic and therapeutic targets for breast cancer.

## 1. Introduction

The homeostasis of extracellular matrix proteins (ECMs) is indispensable for tissue development, wound healing, and normal organ homeostasis. Therefore, the persistent dysregulation of ECMs can result in life-threatening pathologies [1]. Periostin (Pn), one of the ECMs belonging to the fasciclin family, is associated with many fundamental biological processes such as angiogenesis, cell proliferation, cell migration, and collagen assembly in both physiological and pathological processes [2]. In mice and humans, Pn undergoes alternative splicing in its c-terminal region, which lacks known functional domains [3]. We and other groups have observed the differential expression of specific Pn alternative splicing variants (ASVs) in malignancies [4] and cardiovascular disease [5], and their expression levels correlated well with cancer progression and the degree of heart failure. Additionally, the suppression of specific Pn-ASVs ameliorated cancer progression and heart failure [6,7]. Thus, disease-specific Pn-ASVs seem to be potential diagnostic and therapeutic stratification targets. However, the precise distribution pattern of Pn-ASVs under physiological and pathological states is largely unknown. In this study, we address this subject through the use of variant-specific primers for RT-PCR, probes for in situ hybridization (ISH), and antibodies for immune-precipitation (IP) and immune-histochemical (IHC) examinations. The analysis of the expression of Pn-ASVs in normal mouse and human organs demonstrates that digestive organs and skin exhibit high Pn expression levels, where Pn-ASVs lacking exon 17 and 21 comprised a majority of the physiologically expressed Pn. Pn-ASVs with exon 17 or 21 are minimally expressed in normal mouse and human tissue. However, Pn-ASVs with exon 21 present significantly increased expression in tumor but not normal adjacent tissue (NAT) in breast cancer. ISH analysis confirms that stroma cells surrounding cancer cells are the main source for the synthesis of Pn-ASVs with exon 21 in several types of cancers. An in vivo bio-distribution study using ^89^Zr labeling Pn Ab against exon 21 (^89^Zr Pn-21Ab) in tumor-bearing mice confirmed the selective accumulation of the Ab in tumor tissue. Additionally, Pn Ab against exon 21 (Pn-21Ab) significantly suppressed tumor growth in a mouse triple-negative breast cancer model. These data suggest that disease-specific Pn-ASVs, which are rarely expressed in normal tissue, may be attractive targets in terms of their diagnostic and therapeutic potential.

## 2. Materials and Methods

### 2.1. Antibodies and Reagents

In order to develop the monoclonal Ab against exon 14, 17, and 21 of human Pn, the exon 14, 17, and 21 peptides were synthesized, and monoclonal Ab was generated in immunized rat (exon 14) or mouse (exon 17 and 21) models at Immuno-Biological Laboratories Co., Ltd. (Gunma, Japan). Mouse monoclonal Pn Ab against exon 12 was obtained from AdipoGen Life Sciences (San Diego, CA, USA, cat # AG-20B-0033-C100). The specificity of the Ab was tested through dot plot or ELISA testing using peptides from each exon. Wheat Germ Agglutinin (WGA) (iFluor 647) was purchased from AAT Bioquest (Pleasanton, CA, USA, catalog # 25559). pFC14K POSTN v1 Vector (halo-tagged full-length Pn vector) was purchased from Promega (Madison, WI, USA, catalog # 191466G1D).

### 2.2. Human Samples

Human adult normal tissue-specific total protein was obtained from Cosmo Bio Co., Ltd. (Tokyo, Japan). Tissue microarrays of normal human tissue and breast cancers were obtained from US Biomax (Rockville, MD, USA). Use of the collection of human breast cancer tissues was approved by Osaka University Graduate School of Medicine (approved number: 21084). Informed consent was obtained from participants, according to the relevant guidelines of the institutional review board. The patient information is listed in Table 1. In addition, we referred to the human tissue protein expression information provided by the Human Protein Atlas (www.proteinatlas.org (accessed on 10 November 2023)) [8].

### 2.3. Mouse Samples

All mouse experiments were approved by the Institutional Animal Committee at the Department of Veterinary Science of Osaka University Graduate School of Medicine (approved number: 04-101-000) and followed the recommendations of the guidelines for animal experimentation at research institutes (Ministry of Education, Culture, Sports, Science and Technology, Tokyo, Japan), guidelines for proper conduct for animal experimentation (Science Council of Japan, Tokyo, Japan), guidelines for animal experimentation at institutes (Ministry of Health, Labor and Welfare, Chiyoda-ku, Tokyo, Japan), and the ARRIVE guidelines. C57BL6J mice at the age of 8 weeks were used for mPn-ASVs expression analysis. Pn KO mice (B6J strain) were obtained from Charles River (Wilmington, MA, USA), and each of the organs were isolated as negative controls for mPn-ASVs analysis. Postn-tdTomato lineage tracing mice were established by crossing B6.129S-Postn^tm2.1 (cre/Esr1*)^ J mice and B6. Cg-Gt (ROSA) 26Sor^tm14 (CAG-tdTomato) Hze^/J, both obtained from the Jackson laboratory (Bar Harbor, ME, USA). Eight-week-old Postn-tdTomato lineage tracing mice were treated with intraperitoneal injection of tamoxifen for 5 days, and the stomach, small intestine, colon, skin, lung, heart, liver, kidney and cerebrum were isolated for the immunofluorescence study.

### 2.4. Cell Culture

Mouse triple-negative breast cancer (TNBC) cells (4T1 cells) were obtained from ATCC (Manassas, VA, USA), and 4T07 cells were kindly donated by Prof. Saya of Keio University. Both cell lines were cultured in DMEM with 10% fetal bovine serum.

### 2.5. Histological Analysis

Human breast cancer tissue microarray (TMA) samples from US Biomax, Inc. (Derwood, MD, USA), was used for IHC analysis. IHC was performed with a Histofine simple stain MAX PO kit (Nichirei, Tokyo, Japan), according to the manufacturer’s instructions. Organs from Postn-tdTomato lineage tracing mice were fixed in fresh 4% paraformaldehyde for 1 h. Fixed tissues were placed in 10, 20, or 30% sucrose in PBS for dehydration until tissue sinks (6–12 h). Dehydrated tissue was frozen with OCT compound (cat # 4583, Fisher Scientific, Waltham, MA, USA) and then cut into 10 μm sections for fluorescent IHC analysis.

### 2.6. In Situ Hybridization

Human breast cancer TMA samples from US Biomax, Inc. (Derwood, MD, USA), were used for ISH analysis. ISH was performed with prove against Pn, RNAscope^®^ (cat # 409181), or BaseScope^®^ (custom probe), obtained from ACD Bio (Newark, CA, USA) at Advantech Inc (Hyogo, Japan), according to the manufacturer’s instructions. PPIB (positive probe) and DapB (negative prove) were used to check the quality of the tissue sections.

### 2.7. Quantitative Real-Time PCR

Total RNA from tissues of C57BL6J mouse organs was obtained for reverse transcription, as described previously [4]. Briefly, tissues were homogenized, and total RNA was isolated from tissues using the Fast gene RNA premium kit (Nippon genetics, Tokyo, Japan, cat # FG-81250) with QIAzol lysis reagent (QIAGEN, Venlo, The Netherland, cat # 79306). RNA was quantified, and its integrity was confirmed using Nanodrop. A High-Capacity cDNA Reverse Transcription Kit with RNase Inhibitor (ThermoFisher Scientific, Waltham, MA, USA, cat # 4374966) was used to synthesize cDNA from 1 μg total RNA, and an Applied Biosystems QuantStudio 7 (ThermoFisher Scientific, Waltham, MA, USA) was used for detection of target gene expression in accordance with the manufacturer’s instructions. For human breast cancer tissue, freshly isolated tumor and normal mammary tissue contralateral to the tumor across the nipple were collected from a sample of breast cancer patients who had undergone total mastectomy. Total RNA was isolated by FastGene RNA Premium Kit with QIAzol lysis reagent, and 250 ng total RNA was subjected to synthesizing cDNA using SuperScript™ IV First-Strand Synthesis System (Invitrogen™, Waltham, MA, USA, cat # 18091050). Mouse 18 s rRNA primers were amplified as a reference standard. Copy number quantification was performed using specific primer sets and standard mouse and human Pn-ASV plasmids. The used primers are listed in Table 2.

### 2.8. Western Blot Analysis

For Western blotting, total lysates from tissues of C57BL6J mouse organs or human tissue samples were prepared as described previously [4]. Mouse organs and breast cancer tissues were homogenized, and lysates were prepared with RIPA buffer, which were electrophoresed and blotted onto PVDF membranes. For IP study, the indicated number of proteins were precipitated with Pn antibodies (exon 14, 17, and 21) or IgG, and IP proteins were collected using an immunoprecipitation kit (Dynabeads™ Protein G) according to the manufacturer’s instructions (Invitrogen, Tokyo, Japan, cat # 10007D). The blotted membranes were incubated with primary antibodies against Pn exon 12 Ab (AdipoGen Life Sciences Inc., San Diego, CA, USA, cat # AG-20B-0033-C100) and GAPDH (FUJIFILM, Tokyo, Japan, cat # 015-25473) for ECL analysis. Western blot original images can be found in Appendix A.

### 2.9. Establishment of Syngeneic Mouse Tumor Models for In Vivo Bio Distribution Study of Pn Exon 21 Ab

Female (BALB/c, 8–12 weeks old) were subcutaneously injected with 0.1 mL PBS containing 4T07 (1 × 10^6^ cells/mL) cells into the right flank. The appearance and size of tumors, as well as the body weight of the mice, were measured twice a week. The length and width of the tumors were measured using a caliper. Tumor size was calculated by the formula (L × W^2^)/2. When tumors were greater than 100 mm^3^ in size, tumor-bearing mice together with age-matched, wild-type mice were randomized into the respective Pn-Ab treatment arms for the following mouse pharmacokinetic study. 4T07 cells and 4T1 cells are murine breast cancer cell lines, and 4T07 cells express all mPn-ASVs (Appendix A) but, unlike 4T1 cells, are known not to metastasize. Therefore, 4T07 cells are considered suitable for Ab distribution studies. Additionally, to confirm the anti-tumor efficacy of Pn-Ab, female BALB/c mice bearing 4T07 cells were treated with intravenous injection of Pn-21Ab (10 mg/kg or 30 mg/kg), Pn-7/8Ab (10 mg/kg), or saline 2 times per week, and the tumor size was measured twice a week. When mice were sacrificed, serum was collected and mixed from 5 mice in each group, and several cytokines and chemokines were comprehensively examined using the Proteome Profiler Mouse XL Cytokine Array (R&D systems, Minnneapolis, MN, USA, cat # ARY028) to investigate the mechanism of drug efficacy of Pn-21Ab on breast cancer.

### 2.10. Mouse Bio-Distribution Study of ^89^Zr-Labeled Pn-Ab

All labeling and animal experiments were conducted at Nihon Medi-Physics Co., Ltd., Tokyo, Japan. The ^89^Zr labeling method using deferoxamine was employed to obtain ^89^Zr-labeled Pn-21Ab and Pn-7/8Ab with radioactivity suitable for use in animal experiments. The Ab concentrations of ^89^Zr-labeled Pn-21Ab and Pn-7/8Ab were evaluated using a variable optical path length UV–visible spectrophotometer (SoloVPE), and the radiochemical purity up to 2 days after production was evaluated via Thin-Layer Chromatography (TLC). It was confirmed that more than 98% radiochemical purity was maintained up to 2 days after production. Tumor-bearing and normal mice (*n* = 3 each) were injected with ^89^Zr-labeled Pn-21Ab and Pn-7/8Ab intravenously into the tail vein under awake conditions. PET and CT imaging were performed under isoflurane anesthesia at 6, 24, and 72 h after administration, for 10 min each. PET and CT imaging were performed using a Si78 (Bruker, Billerica, MA, USA) small-animal PET/CT system. The ^89^Zr accumulation in tumors and trunks was obtained from composite PET and CT images and analyzed from transverse and coronal views using the PMOD software version 4.4 (PMOD Technologies, Fällanden, Switzerland). In the image analysis, the locations of the tumor, muscle, heart, liver, and kidney were identified from the combined PET and CT images, and a three-dimensional volume of interest (VOI) was established. The standardized uptake value (SUV) was calculated from the VOI at each time point and in each tissue according to the following formula:SUV = radioactivity concentration in the VOI (MBq/mL)/administered radioactivity (MBq)/mouse body weight (g).

Another set of tumor-bearing and normal mice (*n* = 5 each) were injected with ^89^Zr-labeled Pn-21Ab and Pn-7/8Ab. At 72 h after administration, the animals were euthanized through cardiac blood collection under deep anesthesia with isoflurane inhalation (1.0–4.0%). Their organs were collected, and feces and urine were collected from the metabolic cage. The organs analyzed included the heart, lung, spleen, pancreas, stomach, small intestine, colon, uterus, muscle, bone, tumor, liver, kidney, and the rest of the body. The weight (excluding feces and urine) and radioactivity of collected blood and organs were measured, and the radioactivity distribution was evaluated. The radioactivity (count rate) obtained with a γ-ray well scintillation device was time-corrected to the date and time of ^89^Zr administration, and the %ID in all tissues and %ID/g in organs (excluding feces and urine) were calculated.

### 2.11. Statistics

For statistical analysis, the values are presented as the means ± SE. All statistical analyses were performed with the EZR [9] plugin for R. For statistical analysis of expression change between the two groups, unpaired t-test for normally distributed data, and Mann–Whitney U-test for non-normally distributed data were performed. A one-way ANOVA for multiple comparisons and post-hoc Tukey’s HSD for pairwise comparison were carried out. A *p*-value less than 0.05 was considered as statistically significant.

## 3. Results

### 3.1. Periostin Splice Variant Expression Pattern in Adult Mice

Four ASVs of mouse Pn (mPn) and eight for human Pn (hPn) have been reported (Appendix A) [6]. To examine the physiologically expressed mPn-ASVs, several organs were isolated from 8-week-old healthy C57BL6 mice and analyzed. First, the mPn-ASV mRNA expression levels in various organs were investigated using the specific primer sets for mPn-ASVs detailed in Table 1. The copy number for each of the mPn-ASVs was measured through real-time PCR. The mPn-ASVs (mPn 1–4) are classified depending on the inclusion or exclusion of two exons: exon 17 and 21 (Appendix A). As shown in Figure 1A, the expression of mPn-ASVs in each organ was almost restricted to mPn 4, which lacks both exon 17 and 21. The mPn mRNA copy number was relatively high in the lung, stomach, colon, heart, and skin. Next, the expression of all mPn proteins in adult mice was examined using an Ab against Pn exon 12, which theoretically recognizes all Pn-ASVs. As a result, we identified high mPn protein expression in the stomach, colon, and skin; moderate expression in the small intestine, lung, liver, spleen, ovary, heart, and skeletal muscle; and low expression in the cerebrum, cerebellum, and kidney (Figure 1B,C). The expression of total mPn in several organs was also confirmed using three healthy wild-type mice with positive and negative controls. Supernatants from HEK293 cells transfected with plasmids with human Pn 1-Halotag and 4T1 mouse triple-negative breast cancer (TNBC) cell line, which highly expressed all mPn-ASVs, were used as the positive control, while organs from Pn knockout mice (Pn KO) were used as the negative control. As shown in Figure 2A and Appendix A, physiologically expressed mPn in the lung, stomach, colon, and skin showed a similar expression pattern, while mPn from the 4T1 cell culture supernatant demonstrated bands with higher molecular weights than that from wild-type mice, suggesting that mPn variants expressed physiologically are short isoforms. With the aim of analyzing which variants are expressed physiologically at the protein level, immunoprecipitation (IP) and Western blotting were conducted. IP was performed with Ab against exon 14, 17, and 21 following immunoblotting detected with Ab to exon 12. This method allows us to detect all mPn variants on lane 2 (Figure 2B), mPn-ASVs with exon 17 on lane 3, and those with exon 21 on lane 4. As a result, the strongest band was detected on lane 2. Exon 17 Ab could not precipitate any protein, and mPn with exon 21 was slightly detectable in normal skin tissue (Figure 2B circle, Figure 2C dose-dependent expression) but not in the lung and colon (Appendix A). Exon 12 Ab detected similar multiple bands in proteins extracted from organs (lane 1, without IP) to IP proteins with exon 14 Ab (lane 2), as shown in Figure 2B and Appendix A. These data indicate that mPn-ASVs lacking exon 17 and 21 are the most physiologically expressed ASVs in mice. This finding is compatible with the observation of mPn-ASV mRNA expression (Figure 1A). As lanes 1 and 2 showed multiple bands, there must be other physiological mPn-ASVs, other than mPn 4. The distribution of mPn was investigated through the generation of Postn-tdTomato lineage tracing mice. In the presence of tamoxifen, all mPn-ASV-expressing cells permanently express tdTomato. Eight-week-old Postn-tdTomato lineage tracing mice were treated with tamoxifen for 5 days, and the stomach, small intestine, colon, and skin showed increased mPn expression, as depicted in Figure 1 and Figure 2, and the lung, heart, liver, kidney and cerebrum that indicated middle-low mPn expression were isolated. Frozen thin sections were prepared, and mPn expression was observed using a fluorescence microscope. As shown in Figure 3, the mPn expression was confirmed as tdTomato-positive cells in several organs. mPn-positive cells in mucosa (arrows in low magnitude and arrowheads in high magnitude) of the stomach and mucosa (arrows) and submucosa (arrowheads) of the small intestine were observed. In the colon, mPn-positive cells were observed to present a linear peri-cryptal pattern in normal colonic mucosa (arrows) and submucosa (arrowheads). In the skin, mPn-positive cells were localized in the bulge region of the hair follicle (arrows) and at the basement membrane in the epidermis (arrowheads). In the lung, the especially peribronchus area was positive for tdTomato. In the heart, only a small number of cells between myocytes were positive for tdTomato. tdTomato-positive cells were rarely seen in the liver, kidney, and cerebrum. These findings correlated well with the expression intensity observed in Figure 1. Taken together, several organs in mice presented a physiological expression of mPn, and mPn-ASVs lacking exon 17 and 21 were predominantly expressed under physiological conditions.

### 3.2. Periostin Splice Variant Expression Patterns in Normal Human Tissues

Next, we examined the expression levels of hPn-ASVs in several human organs. According to the Human Protein Atlas (www.proteinatlas.org (accessed on 10 November 2023)), the stomach, colon, and lung—which showed physiological high mPn expression in mice—presented similarly high hPn protein expression in human tissues [8]. Furthermore, the breast is also known to present high hPn expression under physiological conditions, while the small intestine and skin are classified as low-expression organs in humans. A Pn Ab (Atlas Antibodies, Sigma-Aldrich, St. Louis, MO, USA, HPA012306) that recognizes the n-terminal region of hPn was used for IHC analysis. This Ab can theoretically detect all hPn-ASVs. As shown in Figure 4A, strong hPn expression was observed in the stomach, colon, and breast. In the lung and skin, the presence of hPn was observed in the alveolar epithelium (arrowheads), dermis layer (arrows), and in the bulge region of the hair follicle (arrowheads). The signal was almost negligible in the spleen (Figure 4A). Then, IP proteins with Ab against Pn exons 14, 17, and 21 of human organs were detected via Western blotting with exon 12 Ab. As shown in Figure 4B, IP protein with Pn exon 14 Ab was highly detectable in the stomach, colon, lung, and breast compared to that with exon 17 or 21 Ab. hPn with exon 21 was faintly detected in lung and breast tissues, as indicated by circles. Exon 12 antibodies detected similar bands in proteins extracted from organs (lane 1, without IP) to IP proteins with exon 14 antibodies (lane 2). Again, these observations indicated that hPn-ASV protein lacking exon 17 and 21 are mainly expressed under physiological conditions in humans, similar to that observed in mice.

### 3.3. Disease-Specific ASVs in Human Breast Cancer Tissue

Our previous studies demonstrated that the suppression of Pn-ASVs with exon 21 overcame chemotherapy resistance in a mouse breast cancer model [6]. Additionally, Pn-21Ab inhibited tumor growth in a manner accompanied by the modulation of the tumor microenvironment (TME) in a mouse xenograft model [4]. To investigate the disease-specific variants of hPn, we examined the expression of hPn-ASVs in breast cancer tissues. First, mRNA expression was examined through real-time PCR using sets of primers designed to amplify segments of cDNA that span from exon 3 to 4 at the n-terminal region in order to detect all ASVs; from exon 16 to 17 for ASVs with exon 17; and exon 21 to 22 for ASVs with exon 21. Breast cancer tissue and NAT (normal adjacent tissue on the opposite side of the nipple from the cancer) were collected from patients who had undergone total mastectomy for localized breast cancer. In 10 out of 11 cases, total hPn mRNA expression was up-regulated in tumor tissue, compared to NAT (Figure 5A). On average, there was a significant increase in the total hPn mRNA expression in tumor tissue, when compared to NAT (Figure 5C). Similarly, the expression of hPn-ASVs with exon 21 mRNA was also increased in tumor tissue, as compared to NAT, in a similar manner as total hPn (Figure 5B,C). Additionally, exon 17 containing hPn-ASV mRNA expression was increased but lacked statistical significance (Figure 5C). These data suggest that hPn-ASV mRNA with exon 21 are predominantly elevated in breast cancer tissue. Next, we assessed hPn protein expression in breast cancer and NAT through Western blotting using Ab against Pn exon 12 and 21. In both tests, the bands were strongly detected in the breast cancer tissues, when compared to NAT (Figure 5D). The ratio of Pn expression in breast cancer tissue to NAT was higher in hPn-ASVs with exon 21 than all hPn-ASVs by 1.6–5.2-fold in four patients and lower in two patients (by 0.5- and 0.8-fold, respectively; Figure 5E). These results again indicate that hPn-ASV protein with exon 21 was predominantly elevated in breast cancer tissue, being detected in tumor tissue with greater sensitivity than total hPn in some cases. To examine the localization of hPn-ASVs with exon 21, ISH was performed. Probes hybridizing to exons 1–8 (RNAscope^®^, Figure 6A,B) and exon 21 (BaseScope^®^, Figure 6C,D) were utilized to confirm the localization of all hPn-ASVs and hPn-ASVs with exon 21, respectively. As shown in Figure 6, the probes were mainly accumulated in the peri-tumoral stroma cells with spindle-shaped nuclei, suggesting that Pn-ASVs with exon 21 are mainly synthesized by stromal cells—possibly cancer-associated fibroblasts (CAFs) in the peri-tumoral region of breast cancer. A similar distribution pattern of hPn-ASV expression in the stroma has been observed in many cancer types, including ovarian and laryngeal cancer (Appendix A). Interestingly, recent reports have suggested that the hPn-positive CAF sub-population cooperates with other cell population to determine the specific TME and patient prognosis [10,11]. Thus, these observations indicate that the hPn-ASVs with exon-21-positive cells in stroma might contribute to the development of a micro-environment that is supportive of cancer cells.

### 3.4. In Vivo Bio Distribution Study of Pn-21Ab

To assess the significance of Pn-ASVs with exon 21 as a diagnostic and therapeutic target, mice bearing 4T07 mouse TNBC tumors were injected intravenously with a single dose of ^89^Zr labeling Pn exon 21 Ab (^89^Zr Pn-21Ab) and exon 7/8 Ab (^89^Zr Pn-7/8Ab) (10 mg/kg/body). 4T07 mouse TNBC tumors expressed both Pn-ASVs with or without exon 21 (Appendix A). Unlike 4T1 TNBC, the 4T07 TNBC syngeneic mouse breast cancer model rarely develops metastasis, thus making it suitable for in vivo Ab distribution assays. One group of animals (*n* = 5 each for diseased mice and non-diseased wild-type mice) were selected for imaging with PET-CT at 6, 24, and 72 h after injection with antibodies. From the composite PET and CT images, ^89^Zr accumulation in the tumor and trunk was assessed in transverse and coronal views through image analysis. In the image analysis, the locations of the tumor (only in diseased mice), muscle, heart, liver, and kidney were identified, and a three-dimensional volume of interest (VOI) was established. A standardized uptake value (SUV) was calculated from the VOI set at each time point and in each tissue according to the formula mentioned in the Methods section. Another set of animals were sacrificed 72 h after injection. Tumors, blood, and other organs of interest were collected and weighed, and their radioactivity was measured. ^89^Z Pn-Ab uptake in each organ is expressed as the percentage of injected dose per gram of tissue (% ID/g) (*n* = 3). PET images revealed clear tumor uptake of both ^89^Zr Pn-21Ab and ^89^Zr Pn-7/8Ab from 6 h to 72 h after intravenous injection (Figure 7A). After tracer injection, the blood levels of ^89^Zr decreased in both antibodies over time, with faster disappearance observed for ^89^Zr Pn-21Ab, and tumor levels increased with higher mean SUV for ^89^Zr Pn-21Ab, resulting in maximal tumor-to-blood ratios of 2.2 ± 0.3 and 1.1 ± 0.3 at 72 h for ^89^Zr Pn-21Ab and ^89^Zr Pn-7/8Ab, respectively (Figure 7B). As the time point for PET imaging with ^89^Zr Pn-Ab presenting the highest tumor-to-blood ratio in 4T07 xenograft mice was 72 h post-injection, the ex vivo bio-distribution at 72 h after injection into mice bearing 4T07 tumor and wild-type mice were examined (Figure 7C,D). Similar uptake in the tumor for both antibodies, lower bio-distribution of ^89^Zr Pn-21Ab in non-tumor tissue (e.g., blood, heart, lung, spleen, small intestine, colon, and skeletal muscle), and higher bio-distribution of ^89^Zr Pn-21Ab in the liver was confirmed, reflecting a higher clearance rate and tumor-to-blood ratio for ^89^Zr Pn-21Ab (Figure 7C). A comparison of the ^89^Zr Pn-21Ab distribution in wild-type and diseased mice revealed that ^89^Zr Pn-21Ab concentrations in each organ tended to be lower in diseased mice, possibly reflecting its accumulation in tumors (Figure 7D). Indeed, when the same doses of Pn-7/8Ab and Pn-21Ab were administered to the 4T07 mouse breast cancer model, Pn-21Ab was found to have a greater anti-tumor effect, as compared to Pn-7/8Ab (Figure 7E). Additionally, there was a dose-dependent trend toward tumor shrinkage in Pn-21Ab-treated mice with 4T07 breast cancer (Figure 7F). The effect of Pn-21Ab on breast cancer was examined by a comprehensive protein analysis using serum from mouse 4T07 breast cancer model. As shown in Appendix A, the expression of angiogenic factors (e.g., Angiopoetinn-1, FGF acidic), chemokines (e.g., CCL2, CXCL13), and inflammatory cytokines (e.g., IL-6, IL-1 beta), which were elevated in serum from tumor embedded mice, was suppressed by Pn-21Ab administration. These results were consistent with the angiogenic and inflammation-inducing effects of Pn-ASVs with exon 21 reported previously [4,7,12,13]. All together, these results support the target-specific accumulation of ^89^Zr Pn-21Ab in the 4T07 xenograft model, indicating the significant diagnostic and therapeutic potential of Pn-ASVs with exon 21.

## 4. Discussion

We have previously demonstrated that specific Pn-ASVs are up-regulated under pathogenic conditions, including breast cancer and heart failure [4,5,6]. In breast cancer, Pn-ASVs with exon 21 control the endothelial–mesenchymal transition of cancer cells, macrophage polarization, and tumor angiogenesis [4,6], resulting in chemo-resistance and metastasis. In myocardial infarction, the persistent expression of Pn-ASVs with exon 17 and 21 (i.e., full-length Pn) has been observed, which led to cardiac fibrosis and heart failure following myocardial ischemia [5,7]. Additionally, the inhibition of Pn-ASVs with variant-specific Ab or through the genetic knockdown of Pn ameliorated chemo-resistance, heart failure, and even diabetic retinopathy [5,7,13,14,15]. These observations suggest that Pn-ASVs have significant potential as diagnostic or therapeutic targets. However, without the precise knowledge of the expression of Pn-ASVs in normal tissues, their diagnostic specificity and therapeutic selectivity remain unclear.

In this study, the distributions of Pn-ASVs in organs and their expression levels were assessed in normal mouse and human tissues, as well as human breast cancer specimens derived from mastectomy. The novel findings were as follows: 1. Normal digestive organs, skin, and lung expressed high levels of Pn mRNA and proteins; 2. Physiologically expressed Pn-ASVs were almost restricted to isoforms lacking exons 17 and 21; 3. In breast cancer, total Pn mRNA and protein level were significantly increased, and Pn-ASVs with or without exon 21 expression were amplified, while Pn-ASVs with exon 21 were almost undetectable in NAT; 4. ^89^Zr Pn-21Ab selectively accumulated in the tumor in a syngeneic mouse model. These findings indicate that Pn-ASVs lacking exon 17 and 21 are physiological variants, which additionally increase in the context of breast cancer. In contrast, Pn-ASVs with exon 21 are pathological variants, which appear along with the progression of the disease.

Pn is an ECM protein secreted mainly from cells of mesenchymal origin [16]. It comprises a multi-protein domain including an amino-terminal cysteine-rich EMI domain, four FAS 1 domains, and a carboxyl-terminal hydrophilic domain. It has been reported that the EMI domain of Pn directly binds to fibronectin and interacts with collagen, while the four FAS 1 domains interact with Tenascin-C, BMP-1, and CCN3 [2,12]. However, the significance of the molecular function of the c-terminal of Pn-ASVs is not well understood at present. As mentioned, disease-specific Pn-ASVs are increased under pathogenic conditions, and, in recent years, additional Pn-ASVs have been reported at the c-terminus other than the Pn-ASVs shown in Appendix A [3,17]. Nonetheless, Pn has been repeatedly demonstrated to be associated with survival rate and chemo-resistance in several cancers [18,19,20,21], the degree of heart failure [22], and the severity of asthma [23,24], without mentioning the ASVs. We have also reported the differential function of Pn-ASVs, that Pn-ASVs with exon 21 induce angiogenesis and macrophage polarization [4], Pn-ASVs with exon 17 interact with Wnt 5a, whereas no evidence of these effects in Pn-ASVs lacking exon 17 and 21 was observed [25]. An in silico-based tertiary structural modeling and an examination of the predicted docking interactions of the various Pn-ASVs with fibrotic proteins have been recently published. Due to alternative splicing at the c-terminus, the protein region encoded by exons 1–15 shows structural disparities even though the amino acid sequence is the same. This structural change is expected to result in differences in docking proteins, suggesting that the Pn-ASVs present at the c-terminus may have an effect on the n-terminal side [26]. Definitely a precise understanding of the functions and expression patterns of Pn-ASVs in the c-terminal region is required in order to comprehend their significance during pathogenesis and potential as diagnostic and therapeutic targets.

Gilbert et al. first indicated the concept of mRNA splicing in 1978 [27]. At present, it is widely accepted as a mechanism by which multiple proteins can be produced from a single gene. Surprisingly, more than 95% of human genes have been found to undergo splicing during development and pathogenesis in a tissue-specific or signal transduction-dependent manner [28]. Unfortunately, the number of genes with known functions in ASVs remains quite limited, although they have recently attracted attention as targets for treatment and diagnosis [29,30,31]. In this line, our research may shed light on the significance of Pn-ASVs in the pathogenesis of breast cancer development, although confounding factors such as age [32,33] and other diseases [34,35] that affect Pn expression need to be analyzed.

## 5. Conclusions

This study demonstrated the following: (1) Under physiological (normal) conditions, the digestive organs, skin, and lung expressed high levels of Pn mRNA and protein. (2) Physiologically expressed Pn-ASVs were almost restricted to isoforms lacking exons 17 and 21. (3) In breast cancer, the total Pn mRNA and protein levels were significantly increased, and Pn-ASVs with or without exon 21 expression were amplified, while Pn-ASVs with exon 21 remained almost undetectable in NAT. (4) Pn-21Ab accumulated selectively in the tumor in a syngeneic mouse model. These findings indicate that Pn-ASVs lacking exon 17 and 21 are physiological variants, which are additionally increased in the context of breast cancer. In contrast, Pn-ASVs with exon 21 are pathological variants and appear along with disease progression. Taken together, these data indicate that Pn-ASVs might have potential as diagnostic and therapeutic targets for breast cancer.

## Figures and Tables

**Figure 1 biomolecules-14-01093-f001:**
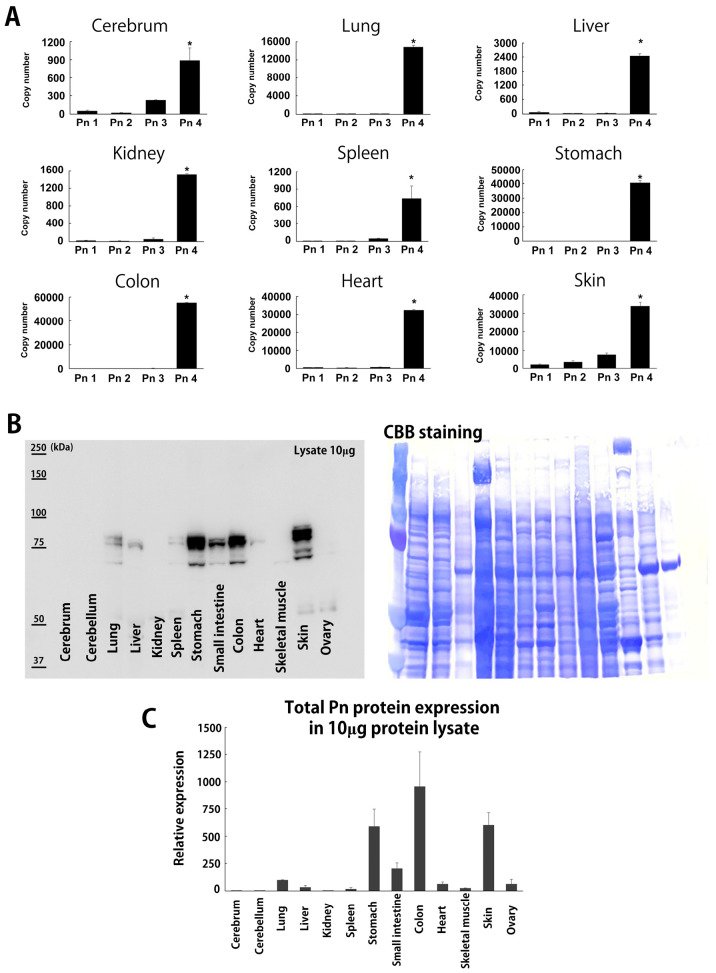
Physiological distribution of Pn-ASV mRNA and total protein in adult mice. (**A**) The mPn-ASV mRNA copy number in several organs from C57BL6J mice aged at 8 weeks. Data are shown as mean ± SE, *n* = 3, * *p* < 0.05 vs. Pn 1–3. An amount of 1 μg total RNA was subjected to cDNA synthesis. The copy number of each Pn-ASV was calculated using specific primer sets and standard mouse Pn-ASV plasmid. (**B**) Total mPn protein expression in several organs from adult mice aged 8 weeks old. A total of 10 μg tissue proteins were loaded and detected with Pn exon 12 Ab, which detected all Pn-ASVs. A membrane with Coomassie brilliant blue (CBB) staining was shown as loading control. (**C**) Quantification data of total mPn protein expression in organs of adult mice.

**Figure 2 biomolecules-14-01093-f002:**
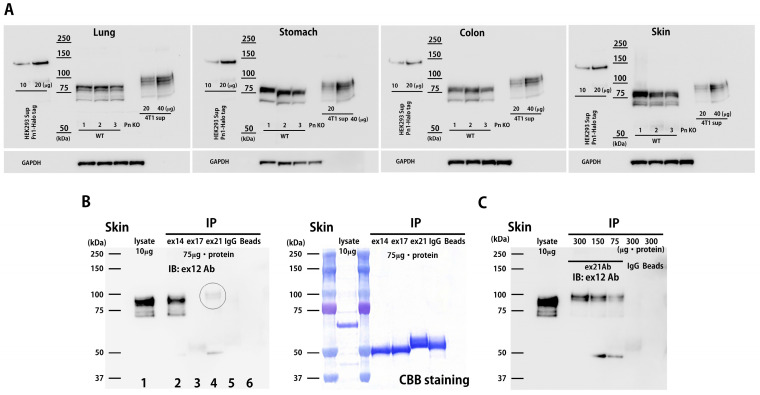
Physiological distribution of mPn-ASV protein in adult mice. (**A**) Confirmation of total mPn protein expression in adult mouse organs. Total mPn protein expression was measured separately in each organ through immunoblotting with Pn exon 12 Ab. Supernatant from human Pn-1-Halotag over-expressed HEK293T cells (10 and 20 μg) and mouse 4T1 breast cancer cells (20 and 40 μg) were simultaneously blotted as a positive control. Proteins from Pn KO mice were used as a negative control. N = 3 for each organ. (**B**) IP and Western blotting results for mPn-ASVs. A total of 75 μg tissue proteins was immunoprecipitated with exon 14 (lane 2), 17 (lane 3), 21 (lane 4), and control IgG (lane 5). Immune complexes were collected on protein G-Agarose beads under agitation, according to the manufacturer’s instructions. Proteins were solubilized in Laemmli buffer, separated via SDS-PAGE, transferred to PVDF membranes, and detected with Pn exon 12 Ab. A total of 10 μg of un-precipitated sample was also loaded in order to determine the molecular level of physiologically expressed mPn-ASVs (lane 1). Lane 6 shows protein precipitated with beads only. The circle indicates the band of mPn-ASVs with exon 21. Coomassie brilliant blue (CBB) staining was performed, according to the manufacturer’s instructions. (**C**) Dosage-dependent increase in IP proteins of mPn-ASVs with exon 21. Amounts of 300, 150, and 75 μg of proteins were immunoprecipitated with Pn exon 21 Ab.

**Figure 3 biomolecules-14-01093-f003:**
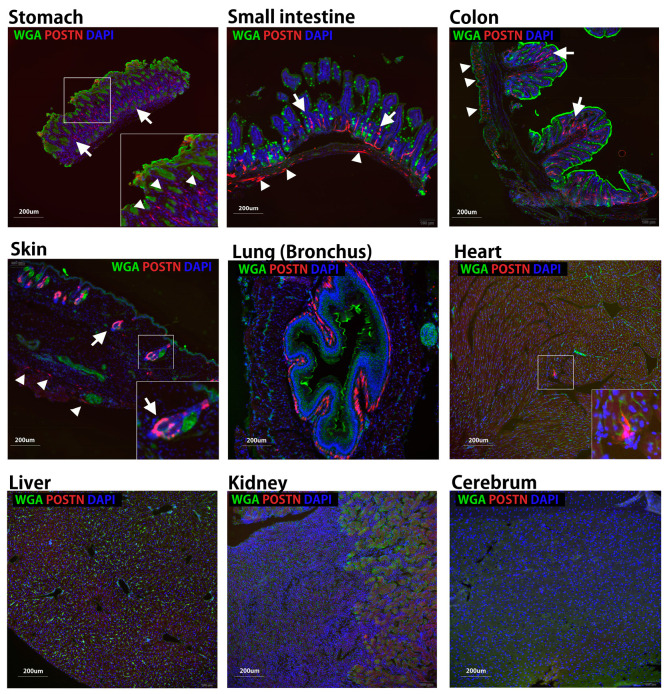
mPn-positive cells in lineage tracing mice. Representative histological sections from several organs of Pn-lineage tracing mice, which show tdTomato-positive labeling cells in the stomach, small intestine, colon, skin, heart, and lung but rarely in the liver, kidney, and cerebrum. In the stomach, interstitial cells in the lamina propria mucosae were positive for tdTomato (arrows in low magnitude and arrowheads in high magnitude), and mucosa (arrows) and submucosa (arrowheads) of the small intestine were observed as positive. In the colon, Pn-positive cells were observed in linear peri-cryptal pattern in normal colonic mucosa (arrows) and submucosa (arrowheads). In the skin, Pn-positive cells were localized in the bulge region of the hair follicle (arrows) and at the basement membrane in the epidermis (arrowheads). In the lung, especially peribronchus area was positive for tdTomato. In the heart, only a small number of cells between myocytes were positive for tdTomato. In the liver, kidney and cerebrum cells expressing tdTomato were rare.

**Figure 4 biomolecules-14-01093-f004:**
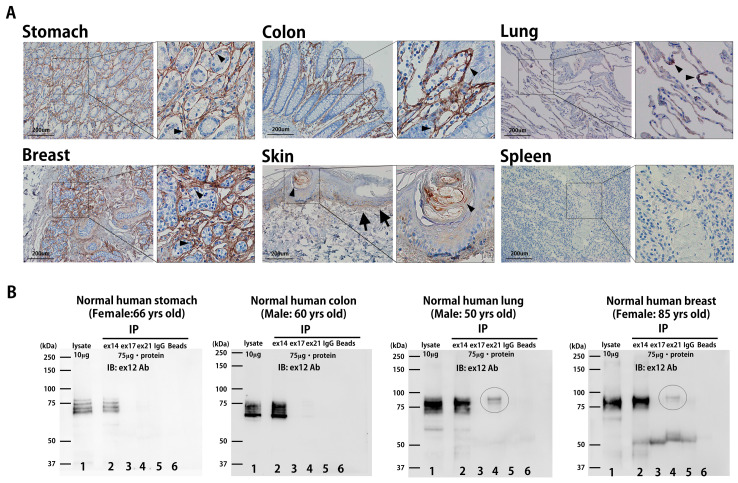
Physiological distribution of hPn-ASV proteins in humans. (**A**) Total hPn expression in normal human organs. Tissue microarrays for normal human tissue were stained with Ab against n-terminal of hPn (Atlas Antibodies, Sigma-Aldrich, HPA012306). (**B**) IP and Western blotting for hPn-ASVs in normal stomach, colon, lung, and breast. A total of 75 μg of tissue proteins were immunoprecipitated with exon 14 (lane 2), 17 (lane 3), 21 (lane 4), and control IgG (lane 5). Immune complexes were collected on protein G-Agarose beads under agitation. Proteins were solubilized in Laemmli buffer, separated via SDS-PAGE, and transferred to PVDF membranes and detected with Pn exon 12 Ab. An amount of 10 μg of un-precipitated sample was also loaded to determine the molecular level of physiologically expressed Pn-ASVs (lane 1). Lane 6 shows beads only. The circle indicates the bands of hPn-ASVs with exon 21.

**Figure 5 biomolecules-14-01093-f005:**
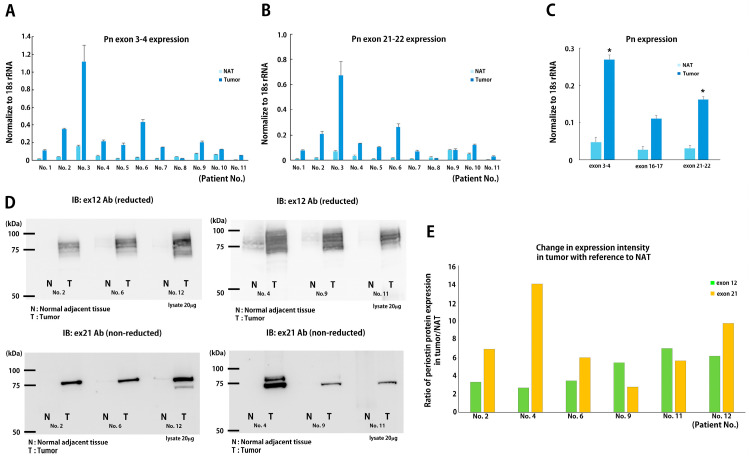
hPn-ASVs expression in human breast cancer and adjacent normal tissue. (**A**) hPn-ASV mRNA expression in breast cancer (tumor) and normal adjacent tissue (NAT). Breast cancer specimens collected from patients undergoing total mastectomy for localized breast cancer were subjected to mRNA analysis. NAT was collected from the contralateral side of tumor across the nipple. Total hPn and hPn-ASVs with exon 17 or 21 were analyzed using specific primer sets and quantitative RT-PCR. Actual values for each patient (**A**,**B**) and average of hPn mRNA expression in tumor and NAT are also shown. (**C**) Data shown as mean ± SE, *n* = 11, * *p* < 0.05 vs. NAT. (**D**) hPn-ASV protein expression in tumor (T) and NAT (N) detected with Ab against exon 12 (all hPn-ASVs, left image) or 21 (right image). Immunoblot with exon 12 or 21 Ab was performed using 10 μg of proteins from 6 patients with breast cancer. (**E**) Quantitative analysis of change in expression in tumor with reference to NAT.

**Figure 6 biomolecules-14-01093-f006:**
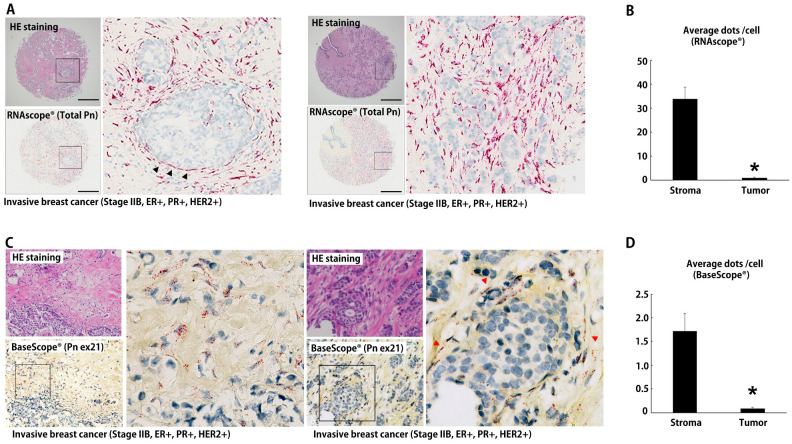
ISH against total hPn and hPn-ASVs with exon 21 in breast cancer. (**A**) ISH with probe for total hPn. Peri-tumoral stroma cells (black triangle) show strong positive signals. Scale bar indicates 500 μm. (**B**) Quantification data of average number of dots per cell. Data shown as mean ± SE, *n* = 22, * *p* < 0.05 vs. stroma (Mann–Whitney U-test). (**C**) ISH with probe for hPn-ASVs with exon 21. Peri-tumoral stroma cells (red triangle) with spindle-shaped nuclei showed strong positive signals. (**D**) Quantification data of average number of dots per cell. Data are shown as mean ± SE, *n* = 22, * *p* < 0.05 vs. stroma (Mann–Whitney U-test).

**Figure 7 biomolecules-14-01093-f007:**
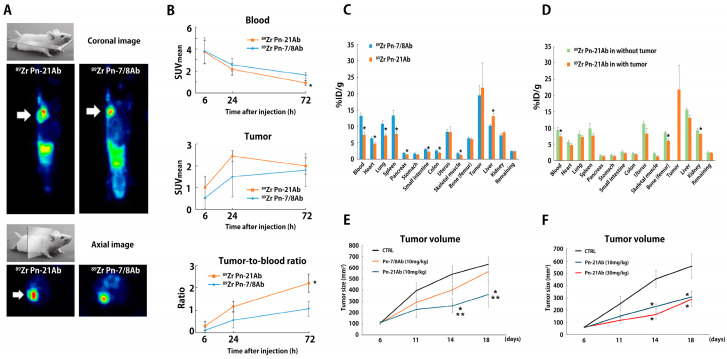
Bio-distribution study of ^89^Zn Pn-21Ab and ^89^Zn Pn-7/8 Ab. ^89^Zr Pn-Ab distribution in BALB/c mice bearing 4T07 mouse TNBC tumor allografts. (**A**) Coronal (upper panel) and axial (lower panel) micro-PET images of BALB/c mice bearing 4T07 allografts (white arrow) 72 h post-^89^Zr Pn-Ab injection. (**B**) Quantification of ^89^Zr Pn-Ab distribution in blood and tumor pool, and tumor-to-blood ratio (*n* = 3). Data shown as median SUVmean ± SE and tumor-to-blood ratio based on SUVmean. * *p* ≤ 0.05 vs. ^89^Zr Pn-21Ab (Mann–Whitney U-test). (**C**) Ex vivo bio-distribution of ^89^Zr Pn-Ab 72 h post-tracer administration (*n* = 5). Data are expressed as %ID/g ± SE. * *p* ≤ 0.05 vs. ^89^Zr Pn-7/8Ab (Mann–Whitney U-test). (**D**) Ex vivo bio-distribution of ^89^Zr Pn-21Ab 72 h post-tracer administration with or without tumor (*n* = 5). Data are expressed as %ID/g ± SE. * *p* ≤ 0.05 vs. ^89^Zr Pn-21Ab in without tumor (Mann–Whitney U-test). (**E**) Pn-ASV-specific Ab (Pn-21Ab and Pn-7/8Ab) inhibited the growth of 4T07 syngeneic mouse model. Mice bearing 4T07 cell tumors were treated with Pn-7/8Ab, Pn-21Ab, or vehicle (*n* = 5 each). The vehicle group received saline, and the other groups were treated with Pn-7/8Ab (10 mg/kg) or Pn-21Ab (10 mg/kg) 2 times per week throughout the experiments starting at day 6. Tumor volumes were recorded as mean ± SE. * *p* < 0.05 vs. CTRL and ** *p* < 0.05 vs. Pn-7/8Ab (Mann–Whitney U test). (**F**) A dose-dependent trend toward tumor shrinkage in Pn-21Ab treated mice with 4T07 breast cancer. Mice bearing 4T07 cell tumors were treated with Pn-21Ab or vehicle (*n* = 5 each). The vehicle group received saline, and the other groups were treated with Pn-21Ab (10 or 30mg/kg), 2 times per week throughout the experiments starting at day 6. Tumor volumes were recorded as mean ± SE. * *p* < 0.05 vs. CTRL (Mann–Whitney U test).

**Table 1 biomolecules-14-01093-t001:** Patient information for breast cancer sample.

Patient No.	Age	Sex	T	N	M	Stage	HG	ER	PgR	HER2
1	61	F	1	1	0	IIA	2	+	-	-
2	53	F	2	2	0	IIIA	2	+	+	-
3	46	F	2	1	0	IIB	1	+	+	-
4	38	F	1	0	0	I	2	+	+	-
5	79	F	2	0	0	IIA	1	-	-	-
6	66	F	1	1	0	IIA	3	-	-	-
7	74	F	1	0	0	I	1	-	-	-
8	77	F	1	0	0	I	3	-	-	-
9	67	F	1	0	0	I	3	-	-	-
10	46	F	1	0	0	I	2	-	-	-
11	56	F	1	0	0	I	3	-	-	-
12	66	F	1	0	0	I	2	+	+	-

Age; sex; TNM classification; cancer stage; histological grade (HG); positive or negative for estrogen receptor (ER), progesterone receptor (PgR), and human epidermal growth factor receptor type 2 (HER2) are shown.

**Table 2 biomolecules-14-01093-t002:** Primer information.

Gene Name	Forward	Reverse
Mouse Pn 1	ATAACCAAAGTCGTGGAACC	TGTCTCCCTGAAGCAGTCTT
Mouse Pn 2	CCATGACTGTCTATAGACCTG	TGTCTCCCTGAAGCAGTCTT
Mouse Pn 3	ATAACCAAAGTCGTGGAACC	TTTGCAGGTGTGTCTTTTTG
Mouse Pn 4	CCCCATGACTGTCTATAGACC	TTCTTTGCAGGTGTGTCTTTT
Mouse 18s rRNA	ATGGCCGTTCTTAGTTGGTG	CGGACATCTAAGGGCATCAC
Human Pn exon 3-4	AAGGAATGAAAGGCTGCCCA	TCCAAGTTGTCCCAAGCCTC
Human Pn exon 16-17	TGTTCGTGGTAGCACCTTCAA	TGATAATAGGCTGAAGACTGCC
Human Pn exon 21-22	GGTCACCAAGGTCACCAAATTC	TGTTGGCTTGCAACTTCCTCAC
Human 18s rRNA	CGGCTACCACATCCAAGGAA	CTGGAATTACCGCGGCT

## Data Availability

The data presented in this study are available on request from the corresponding author.

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
