# Peer review of "Expression of Periostin Alternative Splicing Variants in Normal Tissue and Breast Cancer"

_biomolecules, 2024, doi:10.3390/biom14091093_

Round 1

Reviewer 1 Report

Comments and Suggestions for Authors

Major Comment :

-Might the author clarify how they isolated mRNA from tumor samples. Were the tumor samples fresh or they were FFPE imbedded?  Which reagent did they use to isolate RNA, how much RNA they collect from both tumor and normal tissue? These aspects has to be better clarified in Matheria and Method section.

- Did the authors observe any correlation between the expression level of Periostin or any Pn-ASV and the cancer stage or the aggresivness of the turmo in breast patients?

-The reduction of tumor size in mice trated with Pn-21Ab is not trivial, in my opinion. I guess that this aspect should be enpahsize in the Results section and Supplemmentary Figure 5 should be included into Figure 7. In this regard, may the author investigate more in depth the antitumor impact performed by Pn-21Ab through the analisys of the quality of tumor tissue by Histology (H&E) or investigating possible anti-proliferative or cytotoxic events occurring in tumor cells after treatment in vivo by Immunohistochemistry analisys (testing KI67 or pro-apoptotic molecules such as caspase-3/7 or tunnel assay). This could help to better clarify the possible biological role of exon 21 in breast cancer cells and the therapeutic  impact of Pn-21Ab in breats tumors?

Minor Issue 

-Page 6, lane 275, toypos in the number of exon. Exnon 2 should be corrects with 21. 

Author Response

Reviewer 1

Thank you for reviewer’s accurate point out. Our response to the comment is as follows.

-Might the author clarify how they isolated mRNA from tumor samples. Were the tumor samples fresh or they were FFPE imbedded? Which reagent did they use to isolate RNA, how much RNA they collect from both tumor and normal tissue? These aspects has to be better clarified in Material and Method section.

Answer.

We appreciate this comment. Freshly isolated tumor and normal mammary tissue (contralateral to the tumor across the nipple) were collected from a sample of breast cancer patients who had undergone total mastectomy. The total RNA was collected from the tissue samples using Fast gene RNA premium kit with QIAzol lysis reagent. 250ng of tRNA was used to generate cDNA, which was described in detail in Materials and Methods.

- Did the authors observe any correlation between the expression level of Periostin or any Pn-ASV and the cancer stage or the aggressiveness of the tumor in breast patients?

Answer.

Thank you for the valuable comment. Pn-ASV tended to be higher in cases with lymph node metastasis, but due to the limited number of cases, a clear answer was not obteined. We are currently constructing an ELISA system for the detection of pathological Pn-ASVs, which will make it possible to detect a large number of samples easily in the future.

-The reduction of tumor size in mice treated with Pn-21Ab is not trivial, in my opinion. I guess that this aspect should be emphasize in the Results section and Supplementary Figure 5 should be included into Figure 7. In this regard, may the author investigate more in depth the antitumor impact performed by Pn-21Ab through the analysis of the quality of tumor tissue by Histology (H&E) or investigating possible anti-proliferative or cytotoxic events occurring in tumor cells after treatment in vivo by Immunohistochemistry analisys (testing KI67 or pro-apoptotic molecules such as caspase-3/7 or tunnel assay). This could help to better clarify the possible biological role of exon 21 in breast cancer cells and the therapeutic impact of Pn-21Ab in breats tumors?

Answer.

We appreciate this comment. First, supplement figure 5 has been moved to figure 7. A comprehensive protein analysis of the effect of Pn-21Ab on tumors was performed using serum from mouse 4T07 syngeneic model. The results showed that the expression of angiogenic factors (i.e., Angiopoetinn-1, FGF acidic), various chemokines (i.e.,CCL2, CXCL13), and inflammatory cytokines (i.e.,IL-1b, IL-6) in the blood, which are elevated by tumors, were suppressed by Pn-21Ab administration. These results are consistent with the anti-angiogenic and anti-inflammatory effects of Pn-21Ab reported previously. These results have been added to the manuscript. Also, we are planning to conduct genetic analysis in periostin-positive and negative-cells using Postn-tdTomato lineage tracing mice in a separate study in the future.

Minor Issue

-Page 6, lane 275, toypos in the number of exon. Exnon 2 should be corrects with 21.

Answer.

Thank you for pointing out typos. Using MDPI's English editing system, our manuscript has been reviewed by an experienced native English-speaker.

Reviewer 2 Report

Comments and Suggestions for Authors

The manuscript systematically studied the expression of Periostin gene ASVs in human and mouse in physiological and disease states, and found that Periostin with missing exon 17 and 21 was mainly expressed in physiological state, while Periostin containing exon 21 was mainly expressed in tumor tissue. Further studies in mice found that injection of Periostin antibodies containing exon 21 could inhibit tumor growth. This study will lay the foundation for Periostin ASVs as a target for tumor treatment. However, there are some minor errors in the manuscript, which need to be carefully revised.

1. There are some format errors in the manuscript. For example, line 156 106/mL, line 159 100mm3, line 207-221 and line 377-379 should be deleted.

2. The author described in the article that there are 8 ASVs of human Periostin (Supplemental figure 1), but this data is a bit old. The latest data in the NCBI genebank database shows that Periostin (Gene ID: 10631) has 11 ASVs. Did the author conduct a comparative analysis of these new ASVs? A similar situation also exists in the ASVs of Mouse Periostin. In addition, it is recommended to add the accession number of each ASV.

3. It is recommended to discuss the possible functions of Periostin exon21 or exon 17 in the discussion part.

Comments on the Quality of English Language

The language is acceptable, see "Comments and Suggestions for Authors
" for details.

Author Response

Reviewer 2

Thank you for reviewer’s appropriate comments. Our response to the comments are below.

The manuscript systematically studied the expression of Periostin gene ASVs in human and mouse in physiological and disease states, and found that Periostin with missing exon 17 and 21 was mainly expressed in physiological state, while Periostin containing exon 21 was mainly expressed in tumor tissue. Further studies in mice found that injection of Periostin antibodies containing exon 21 could inhibit tumor growth. This study will lay the foundation for Periostin ASVs as a target for tumor treatment. However, there are some minor errors in the manuscript, which need to be carefully revised.

  1. There are some format errors in the manuscript. For example, line 156 106/mL, line 159 100mm3, line 207-221 and line 377-379 should be deleted.

Answer.

Thank you for pointing out typos and unnecessary sentences. We corrected them accordingly. Also, using MDPI's English editing system, our manuscript has been reviewed by an experienced native English-speaker.

  1. The author described in the article that there are 8 ASVs of human Periostin (Supplemental figure 1), but this data is a bit old. The latest data in the NCBI genebank database shows that Periostin (Gene ID: 10631) has 11 ASVs. Did the author conduct a comparative analysis of these new ASVs? A similar situation also exists in the ASVs of Mouse Periostin. In addition, it is recommended to add the accession number of each ASV.

Answer.

Thank you for your important comments. As reviewer pointed out, it is reported that 11 splicing variants in humans and 6 variants in mice including a variant lacking exon 20. First of all, we mentioned in the Discussion that there are multiple splicing variants other than the variants shown in supplement figure 1. We also mentioned in 3.1.1 that there are other physiological variants besides mouse Pn 4 and human Pn 4-1, since the physiological variants detected by the exon 14 antibody show multiple bands in the healthy organs analyzed by immunoprecipitation assay. In addition, accession numbers are listed in supplement figure 1.

  1. It is recommended to discuss the possible functions of Periostin exon21 or exon 17 in the discussion part.

Answer.

We appreciate this recommendation. First, supplement figure 5 has been moved to figure 7. And a comprehensive protein analysis of the effect of Pn-21Ab on tumors was performed using serum from mouse 4T07 syngeneic model. The results showed that the expression of angiogenic factors (i.e., Angiopoetinn-1, FGF acidic), various chemokines (i.e.,CCL2, CXCL13), and inflammatory cytokines (i.e.,IL-1 beta, IL-6) in the blood, which are elevated by tumors, were suppressed by Pn-21Ab administration. These results are consistent with the angiogenic and inflammation-induced effects of Pn-ASVs with exon 21, which is reported previously. Also we mentioned the possible mechanism of the function of Pn-ASVs with exon 17 in the discussion.

Reviewer 3 Report

Comments and Suggestions for Authors

The authors investigated Pn-ASVs containing exon21 in both normal and breast cancer tissues. Their analysis revealed that Pn-ASVs with exon21 were significantly elevated in tumoral cells compared to adjacent normal tissue in both human and mouse models. Utilizing various methodological approaches, they confirmed this differential expression. Furthermore, the application of Pn Ab targeting exon 21 resulted in the suppression of tumor growth in a mouse model of triple-negative breast cancer. This study is interested and provided a new potential therapeutic target for breast cancer especially the triple-negative breast cancer.

There are some suggestions as below:  

1.        Please move the related figures under the result paragraph.

2.        Authors showed that Pn-ASVs lacked exon 17 and exon 21 in the normal tissues from section 3.1.1 and 3.1.2. However, when they detected exon 3 to 4 for all ASVs, exon 16 to 17 for exon17 absent, and exon 21 to 22 for exon 21 absent, these three fragment mRNA expressions of NAT group look quite similar. Could authors do statistical analysis and put the result in the Fig 5 as well?

3.        Supple Fig 5 should put in the main figures

Comments on the Quality of English Language

Some sentences are too long. Shortening them and dividing them into two to three sentences could enhance readability.

Author Response

Reviewer 3

Thank you for reviewer’s accurate point out. Our response to the comment is as follows.

The authors investigated Pn-ASVs containing exon21 in both normal and breast cancer tissues. Their analysis revealed that Pn-ASVs with exon21 were significantly elevated in tumoral cells compared to adjacent normal tissue in both human and mouse models. Utilizing various methodological approaches, they confirmed this differential expression. Furthermore, the application of Pn Ab targeting exon 21 resulted in the suppression of tumor growth in a mouse model of triple-negative breast cancer. This study is interested and provided a new potential therapeutic target for breast cancer especially the triple-negative breast cancer.

There are some suggestions as below: 

  1. Please move the related figures under the result paragraph.

Answer.

Thank you for the accurate suggestion. We have moved the related figures under the result paragraph. Also, using MDPI's English editing system, our manuscript have been reviewed by an experienced native English-speaker.

  1. Authors showed that Pn-ASVs lacked exon 17 and exon 21 in the normal tissues from section 3.1.1 and 3.1.2. However, when they detected exon 3 to 4 for all ASVs, exon 16 to 17 for exon17 absent, and exon 21 to 22 for exon 21 absent, these three fragment mRNA expressions of NAT group look quite similar. Could authors do statistical analysis and put the result in the Fig 5 as well?

Answer.

Comparing the graphs in Figure 5A and B, Pn 3-4 mRNA expression is higher than Pn 21-22 mRNA expression in NAT in each patient, but taking the average of the all cases analyzed, the difference of Pn 3-4 and Pn 21-22 expression were decreased as shown in figure 5C. Due to the different breast cancer patient backgrounds, it is likely that the breast cancer stage and treatment affected the expression of Pn-ASVs in NAT.

  1. Supple Fig 5 should put in the main figures

Answer.

We appreciate this recommendation. First, supplement figure 5 has been moved to figure 7. A comprehensive protein analysis of the effect of Pn-21Ab on tumors was performed using serum from mouse 4T07 syngeneic model. The results showed that the expression of angiogenic factors (i.e. Angiopoetinn-1, FGF acidic), various chemokines (i.e. CCL2, CXCL13), and inflammatory cytokines (i.e. IL-1b, IL-6) in the blood, which are elevated by tumors, were suppressed by Pn-21Ab administration. These results are consistent with the anti-angiogenic and anti-inflammatory effects of Pn-21Ab reported previously. These results have been added to the paper. Also, we are planning to conduct genetic analysis in periostin-positive and negative-cells using Postn-tdTomato lineage tracing mice in a separate study in the future.

Reviewer 4 Report

Comments and Suggestions for Authors

The manuscript by Kanemoto et al. entitled "Periostin Alternative Splicing Variants Expression in Normal Tissue and Breast Cancer”examined normal tissues and breast cancer tissues focusing on Pn-ASVs expression pattern to proof the significance of pathologically expressed Pn-ASVs as a potential diagnostic and therapeutic targets. Overall the experiments are expertly carried out and the paper is well-written. There are still some suggestions to improve the quality of the manuscript although the findings in this manuscript are potentially interesting.

1. There should be loading control in figure 1B

2. The authors should be careful about the writing. For example, ug should be μg.

3. In figure 3, the authors should do more tissue and compare with the results in figure 4.

4. There should be error bar in figure 5A and 5B.

5. The mechenism of Periostin alternative splicing should be studied.

Comments on the Quality of English Language

could be improved.

Author Response

Reviewer 4

Thank you for reviewer’s accurate point out. Our response to the comment is as follows.

The manuscript by Kanemoto et al. entitled "Periostin Alternative Splicing Variants Expression in Normal Tissue and Breast Cancer”examined normal tissues and breast cancer tissues focusing on Pn-ASVs expression pattern to proof the significance of pathologically expressed Pn-ASVs as a potential diagnostic and therapeutic targets. Overall the experiments are expertly carried out and the paper is well-written. There are still some suggestions to improve the quality of the manuscript although the findings in this manuscript are potentially interesting.

  1. There should be loading control in figure 1B

Answer.

We agree on the opinion. We added CBB staining to figure 1B as loading control.

  1. The authors should be careful about the writing. For example, ug should be μg.

Answer.

Thank you for pointing out typos. Using MDPI's English editing system, our manuscript have been reviewed by an experienced native English-speaker.

  1. In figure 3, the authors should do more tissue and compare with the results in figure

Answer.

We totally agree on the reviewer comment. New tissue images were added to figure 3 including colon, stomach, small intestine, skin, heart, lung, kidney, liver and cerebrum. The expression of periostin traced by tdTomato was compatible with the periositn protein expression shown in figure 1.

  1. There should be error bar in figure 5A and 5B.

Answer.

We appreciate the reviewer comment. We added error bar in figure 5A and 5B.

  1. The mechanism of Periostin alternative splicing should be studied.

Answer.

We appreciate this recommendation. First, supplement figure 5 has been moved to figure 7. And a comprehensive protein analysis of the effect of Pn-21Ab on tumors was performed using serum from mouse 4T07 syngeneic model. The results showed that the expression of angiogenic factors (i.e. Angiopoetinn-1, FGF acidic), various chemokines (i.e. CCL2, CXCL13), and inflammatory cytokines (i.e. IL-1 beta, IL-6) in the blood, which are elevated by tumors, were suppressed by Pn-21Ab administration. These results are consistent with the angiogenic and inflammation-induced effects of Pn-ASVs with exon 21, which is reported previously. Also we mentioned the possible mechanism of the function of Pn-ASVs with exon 17 in the discussion.

Round 2

Reviewer 4 Report

Comments and Suggestions for Authors

It seems that total protein has much different in the Figure 1B CBB straining. The authors should use GAPDH as loading control. Also, there are many bands in the western results of Figure 1B and Figure 2. The authors should show clearly which band produced by which alternative splicing isofrom.

Author Response

Thank you again for the valuable comments. Please see the attachment for the response to the comments. We appreciate the opportunity to improve our manuscript.
